# Biosynthesis of Gamma-Aminobutyric Acid by Engineered *Clostridium tyrobutyricum* Co-Overexpressing Glutamate Decarboxylase and Class I Heat Shock Protein

**Ziyao Liu [1], Xiaolong Guo [1], Kaiqun Dai [1], Jun Feng [1], Tiantian Zhou [1], Hongxin Fu [1,2] and Jufang Wang [1,2,*]**

[1]  School of Bioscience & Bioengineering, South China University of Technology, Guangzhou 510006, China; 202020149006@mail.scut.edu.cn (Z.L.); brucegxl@163.com (X.G.); 202010108563@mail.scut.edu.cn (K.D.); fengjun1399@126.com (J.F.); 201930502453@mail.scut.edu.cn (T.Z.); hongxinfu@scut.edu.cn (H.F.)

[2]  Guangdong Provincial Key Laboratory of Fermentation and Enzyme Engineering, South China University of Technology, Guangzhou 510006, China

*  Correspondence: jufwang@scut.edu.cn

**Abstract:** Gamma-aminobutyric acid (GABA) is a major inhibitory neurotransmitter in the mammalian central nervous system that has a significant beneficial effect on human health. Traditional microbial GABA synthesis requires continuous oxygen supplementation. Here, a new anaerobic platform for GABA production was established with engineered *C. tyrobutyricum* ATCC 25755, which is considered an ideal anaerobic microbial-cell factory for bioproduction. Glutamate decarboxylase (GAD) and Class I heat-shock proteins were screened and overexpressed, generating an excellent Ct-pMAG strain for monosodium-glutamate (MSG) tolerance and GABA production, with a GABA titer of 14.26 g/L in serum bottles with the mixed substrate of glucose and MSG. Fed-batch fermentation was carried out in a 5 L bioreactor, achieving 35.57 g/L and 122.34 g/L final titers of GABA by applying the pH-free strategy and the pH-control strategy, respectively using MSG. Finally, a two-stage strategy (growth stage and bioconversion stage) was applied using glutamate acid (L-Glu) and glucose as the substrate, obtaining a 400.32 g/L final titer of GABA with a productivity of 36.39 g/L/h. Overall, this study provides an anaerobic-fermentation platform for high-level bio-GABA production.

**Keywords:** *C. tyrobutyricum*; GABA; heat-shock protein; biocatalyst





## 1. Introduction

Gamma-aminobutyric acid (GABA), a major inhibitory neurotransmitter in the mammalian central nervous system, has been reported to have hypotensive, diuretic, and sedative effects [1]. GABA is extensively dispersed in animals, plants, and microorganisms, and its benefits for human health are highly regarded [2]. In the field of food additives, Gabaron tea (>150 mg GABA/100 g), a GABA-enriched tea beverage with the ability to decrease blood pressure, was created by Japanese scientists in the 1980s and attracted widespread attention in the market [3,4]. More recently, in 2009, the Chinese Ministry of Health approved GABA for use in the production of beverages, cocoa, chocolate, candies, baked goods, and puffed foods. [3]. Due to numerous health advantages and the commercial value of GABA, there was a rapid rise in demand on the market. In the past, GABA could be synthesized chemically; however, the toxic effects of byproducts and environmental concern severely restricted its use [5,6]. It was difficult to remove and purify the product from the plant material, and the output of GABA enrichment from plants was insufficient to fulfill the market [1]. Microbial synthesis of GABA has recently received great attention due to its benefits of high specificity, environmental friendliness, cost-effectiveness, and numerous generating strains [7–10].

Glutamate decarboxylase (GAD; EC 4.1.1.15) plays a key role in GABA synthesis [11], consuming intracellular protons and converting glutamate to GABA and $CO_2$ [12]. Recently,

numerous microorganisms have been used to produce GABA. Among them, *Escherichia coli*, lactic-acid bacteria (LAB), and *Corynebacterium glutamicum* could serve as the cell-factory chassis [8,13–15]. Synthesizing GABA utilizing whole-cell bioconversion with *E. coli* as the cell biocatalyst showed great potential because of its high efficiency and simple preparation. A total of 308.26 g/L GABA with a molar yield of 99.6% from 3 mol/L L-Glu was achieved with the engineered *E. coli* with a modified cascade system [16]. Using *E. coli*, 4.8 g/L GABA was achieved from glucose with a yield of 49.2% (mol/mol glucose) by applying the metabolic-state-switching method to rewire the metabolic regulatory network [9]. LAB was a competitive GABA producer and mainly contains the genera *Lactobacillus*, *Lactococcus*, *Bifidobacterium*, *Enterococcus*, and *Streptococcus* [17–21]. *Lactobacillus brevis* CRL 2013 was isolated from quinoa and could produce 265 mM GABA with a molar yield of over 99% from modified MRS broth with MSG [22]. A two-stage pH-control strategy was widely applied to GABA fermentation, the parameters of which could be adjusted for bacterial growth or GABA synthesis separately [2]. The glutamate-decarboxylase system was reconstructed in *Lactococcus lactis* and 33.52 g/L GABA was achieved from MSG under a two-stage pH-control strategy [14]. *C. glutamicum* is an important industrial producer of various amino acids, including glutamic acid, which could be further developed for GABA production. A remarkable GABA concentration of 70.6 g/L was achieved using engineered *C. glutamicum* APLGGP from glucose through a two-stage pH-control strategy [8]. Recently, a dynamic metabolic-control method was employed in engineered *C. glutamicum* G7-1, resulting in 45.6 g/L of GABA production from a low-value substrate—glycerol—which exhibited great potential in biorefinery [15].

The traditional bio-based GABA synthesis process usually requires continuous oxygen supplementation for the growth of the microbes. Anaerobic-fermentation technology has advantages over aerobic fermentation, including a simple operation method, cheap operating costs, and better stability [23]. Cell number is a key factor responsible for GABA production [9]; however, anaerobic conditions interfere with the bacterial growth of these traditional aerobic GABA producers, creating obstacles to obtaining a large cell count. Currently, several attempts have been made to carry out the synthesis of GABA under anaerobic conditions, such as using *Levilactobacillus brevis*, *Lactobacillus brevis*, *Bifidobacterium adolescentis*, and *Bifidobacterium dentium*, but their GABA production (4.94–43.99 g/L) was unsatisfactory [19,24,25]. *Clostridium tyrobutyricum* is an anaerobic, Gram-positive, spore-forming, and non-pathogenic bacterium mainly used for short-chain fatty-acid production [26]. It is regarded as an ideal anaerobic-chassis cell and has been engineered to generate multiple chemical compounds, such as n-butanol and butyl butyrate [27,28]. This non-pathogenic strain is favored for industrial use due to environmental health and safety considerations [26,29]. Thus, in this work, a new anaerobic platform for GABA production was established with engineered *C. tyrobutyricum* ATCC 25755 by introducing glutamate decarboxylase since it does not have native glutamate decarboxylase. Considering the stress from the high concentration of substrate, heat-shock proteins were employed to improve strain tolerance and GABA production. Finally, pH-free, pH-control, and two-stage strategies for GABA synthesis were developed in a 5 L bioreactor.

## 2. Materials and Methods

### 2.1. Organism, Plasmids, and Culture Conditions

The sources of bacteria and plasmids used in this study are listed in Table S1. *C. tyrobutyricum* ATCC 25755 was preserved in our lab and anaerobically cultured in Reinforced Clostridial Medium (RCM) or Clostridial Growth Medium (CGM) at 37 °C unless otherwise mentioned [30]. RCM consists of 5 g/L glucose, 5 g/L NaCl, 10 g/L peptone, 10 g/L beef extract, 3 g/L yeast extract, 3 g/L sodium acetate, 1 g/L soluble starch, and 0.5 g/L Cysteine-HCl. CGM consists of 2 g/L yeast extract, 4 g/L peptone, 1 g/L $K_2HPO_4 \cdot 3H_2O$, 0.5 g/L $KH_2PO_4$, 2 g/L $(NH_4)_2SO_4$, 0.1 g/L $MgSO_4 \cdot 7H_2O$, 0.015 g/L $FeSO_4 \cdot 7H_2O$, 0.015 g/L $CaCl_2 \cdot 2H_2O$, 0.01 g/L $MnSO_4 \cdot H_2O$, 0.02 g/L $CoCl_2 \cdot 6H_2O$, and 0.02 g/L $ZnSO_4 \cdot 7H_2O$. *E. coli* CA434 was cultured in Luria–Bertani (LB) medium and

employed to construct and maintain the recombinant plasmids [31]. The medium was sterilized by autoclaving for 20 min at 121 °C. $OD_{600}$ was measured by a 96-well reader (Molecular Devices, San Jose, CA, USA, SpectraMax® M5).

### 2.2. Construction of the Engineered C. tyrobutyricum

GAD genes from *E. coli*, *Lactococcus lactis* JN001, and *Lactobacillus plantarum* JN002 were cloned into plasmid pMTL82151 under the control of the promoter Pcat1 to generate strains names such as pMA01, pMA02, and pMB01, respectively. The Class I HSGs genes *groES* and *groEL* were first amplified from the genomic DNA of *C. tyrobutyricum* ATCC 25755 and then ligated with Pcat1. Then, the fragment was inserted into pMA02 to create plasmid pMAG. Recombinant plasmids were transformed into *C. tyrobutyricum* to generate Ct-pMA01, Ct-pMA02, Ct-pMB01, and Ct-pMAG by conjugation as previously described (Figure S1) [29]. To be brief, cell pellets of *E. coli* CA434 with recombinant plasmids were collected and then mixed with *C. tyrobutyricum*. Then, the mixture was spread on the RCM agar plate for subculture (37 °C for 24 h), followed by screening recombinant *C. tyrobutyricum* on a CGM select agar plate (37 °C for 48–60 h) containing 25 μg/mL thiamphenicol and 250 μg/mL D-cycloserine.

### 2.3. Quantitative Real-Time PCR and Relative-Gene-Expression Analysis

As previously reported, the GTP-binding-protein gene was chosen to be the house-keeping gene [29]. Ct-pMA02 was cultured in RCM medium and MSG was added to the treatment group. At the mid-exponential-growth phase of Ct-pMA02, quantitative real-time PCRs were applied to analyze the expression of Class I HSP genes under MSG stress. Strains were treated with two concentrations of MSG (10 g/L or 40 g/L) and samples were collected after MSG treatment for RNA isolation. Samples without MSG treatment were also collected at the same time as the control. RNA extraction, cDNA preparation, and PCR were carried out as previously described [29]. Primers for quantitative real-time PCR are listed in Table S2.

### 2.4. GAD-Enzyme-Activity and Cell-Bound-Activity Analysis

*C. tyrobutyricum* were harvested by centrifugation at 12,000 rpm for 1 min. One mL of 10 OD cell pellets were collected, washed once, and resuspended in 1 mL PBS. Samples were sonicated using an ultrasonic cell crusher (SCIENTZ, Ningbo, China, SCIENTZ-IID) under sonic power of 100 W for 3 s, followed by an interval of 3 s, lasting 10 min. After sonication, the cell extracts were centrifuged at 12,000 rpm for 5 min and the supernatant was collected for the cytoplasmic-GAD-activity determination. As per the previously described method with a few modifications, the activity of the GAD enzyme was determined by measuring the amount of GABA formed at 37 °C in a 500 μL reaction mixture for 10 min, which consisted of 50 μL of cell-extract supernatant, 0.2 mol/L $Na_2HPO_4$-citrate buffer, 20 mmol/L MSG, and 0.5 mmol/L pyridoxal-5-phosphate (PLP) [32]. One unit (U) of cytoplasmic GAD activity was defined as 1 μmol GABA produced in 1 min under the above conditions. Similarly, in order to determine the cell-bound GAD activity, 0.5 mL of 10 OD cell pellets were collected, washed once by PBS, and resuspended in 500 μL reaction mixture. Cell-bound activity was determined in the same reaction system (without cell-extract supernatant) as cytoplasmic GAD activity. One unit ($U/OD_{600}$) of cell-bound GAD activity was defined as 1 μmol of GABA produced in 1 min per OD cell pellets under the above conditions. The reaction mixture was centrifuged at 12,000 rpm for 5 min to collect the supernatant and GABA content was determined by Berthelot reaction [32].

### 2.5. MSG-Tolerance Test

To evaluate the inhibitory effect of MSG on bacterial growth, *C. tyrobutyricum* wild-type and engineered strains were cultured in CGM medium at 37 °C in serum bottles with different concentrations of MSG (10, 20, 30, and 40 g/L). All samples were carried out in

triplicate. Samples were collected every 2 h for 36 h and $OD_{600}$ was measured. The specific growth rate was calculated by analyzing the data as previously described [29].

*2.6. Fermentation Experiments*

The fermentation kinetics of *C. tyrobutyricum* were determined in serum bottles with 50 mL of CGM containing 10 g/L MSG. A total of 24 g/L glucose was added to the medium when the fermentation started. Similarly, the fermentation kinetics of *C. tyrobutyricum* in a 5 L bioreactor were investigated. When pH-free and pH-control strategies were applied, 1 L CGM medium was involved in the bioreactor with initial 30 g/L MSG and 60 g/L glucose. MSG was dissolved in $H_2O$ to prepare 50 mL MSG aliquot (600 g/L) and 200 mL MSG aliquot (700 g/L) as previously described [33]. When the two-stage strategies were applied, 60 g/L, 90 g/L, and 120 g/L glucose were added initially, respectively. Then, 4 mol/L L-Glu was added to generate the bioconversion. $N_2$ was sparged into the medium to generate anaerobic conditions before inoculation. The fermentation was operated at 37 °C with a stirring speed of 150 rpm, and pH was controlled by $NH_3 \cdot H_2O$ or $H_2SO_4$ solution when necessary.

The concentrations of GABA and MSG in the fermentation broth were determined by HPLC (Shimadzu, Kyoto, Japan, LC-20A) with an Agilent zorbax SB C18 column (4.6 mm × 150 mm, 5 μm). Samples were pretreated by precolumn derivatization using 1% 2,4-dinitrofluorobenzene. A binary gradient was elution-program applied with mobile phase A (0.05 mol/L sodium acetate, 1% N, N-dimethylformamide, pH 6.8) and mobile phase B (acetonitrile: $H_2O$ 1:1 by volume). The gradient-elution program was as follows: The flow rate was set to 1 mL/min; equilibration (6 min, 15% B), gradient (16 min 15–30% B, 8 min 30–100% B), cleaning (1 min, 100% B), and equilibration (3 min, 15% B). The column temperature was maintained at 40 °C. Samples were detected at UV 360 nm.

## 3. Results

### 3.1. Amino-Acid-Sequence Analysis and Enzymatic-Property Comparison of Putative GADs

GAD is responsible for acid tolerance and GABA production in many bacteria [34]. The putative *gadB* gene encoding GAD was amplified from the genomic DNA of *L. lactis* JN001 and *L. plantarum* JN002, which were separated from shrimp-aquaculture water and preserved in our laboratory to generate pMA02 and pMB01. DNA sequencing suggested that *lcgadB* from *L. lactis* JN001 consisted of 1401 bases, encoding 466 amino-acid residues with a calculated molecular weight of 54.0 kDa and a pI of 5.02. *LpgadB* from *L. plantarum* JN002 consisted of 1404 bases, encoding 467 amino-acid residues with a calculated molecular weight of 53.4 kDa and a pI of 5.63. BLAST results of the protein-sequence alignment suggest that LcgadB shared 98.93%, 97.64%, and 44.92% homology with the GAD of *L. lactis* CV56, *L. lactis* MG1363, and *E. coli* Nissle 1917, respectively (Figure 1). As for LpgadB, it shared 68.52%, 68.52%, and 44.57% homology with the GAD of *L. lactis* CV56, *L. lactis* MG1363, and *E. coli* Nissle 1917, respectively.

EcgadB, LcgadB, and LpgadB were cloned into *C. tyrobutyricum* and the engineered strains were named Ct-pMA01, Ct-pMA02, and Ct-pMB01, respectively. Cytoplasmic GAD activity was determined to investigate the enzymatic properties of different GADs, and the results suggest that wild-type *C. tyrobutyricum* ATCC 25755 had no glutamate-decarboxylase activity since it appeared as the color of substrate, whereas Ct-pMA02 and Ct-pMB01 turned a deeper blue than Ct-pMA01, indicating that higher GAD activity was achieved by introducing LcgadB and LpgadB (Figure 2A). GAD activity in the exponential phase and the stationary phase was determined using cell extracts of engineered strains under the pH gradient (Figure 2B,C). Overall, the GAD activity of engineered strains was higher in the stationary phase than in the exponential phase, and there was a rapid increase in GAD activity of the three engineered strains from pH 3.5 to 4.5, which then quickly dropped. The activity of Ct-pMA02 and Ct-pMB01 was the highest at pH 4.5, and Ct-pMA02 exhibited overall higher activity than Ct-pMB01.

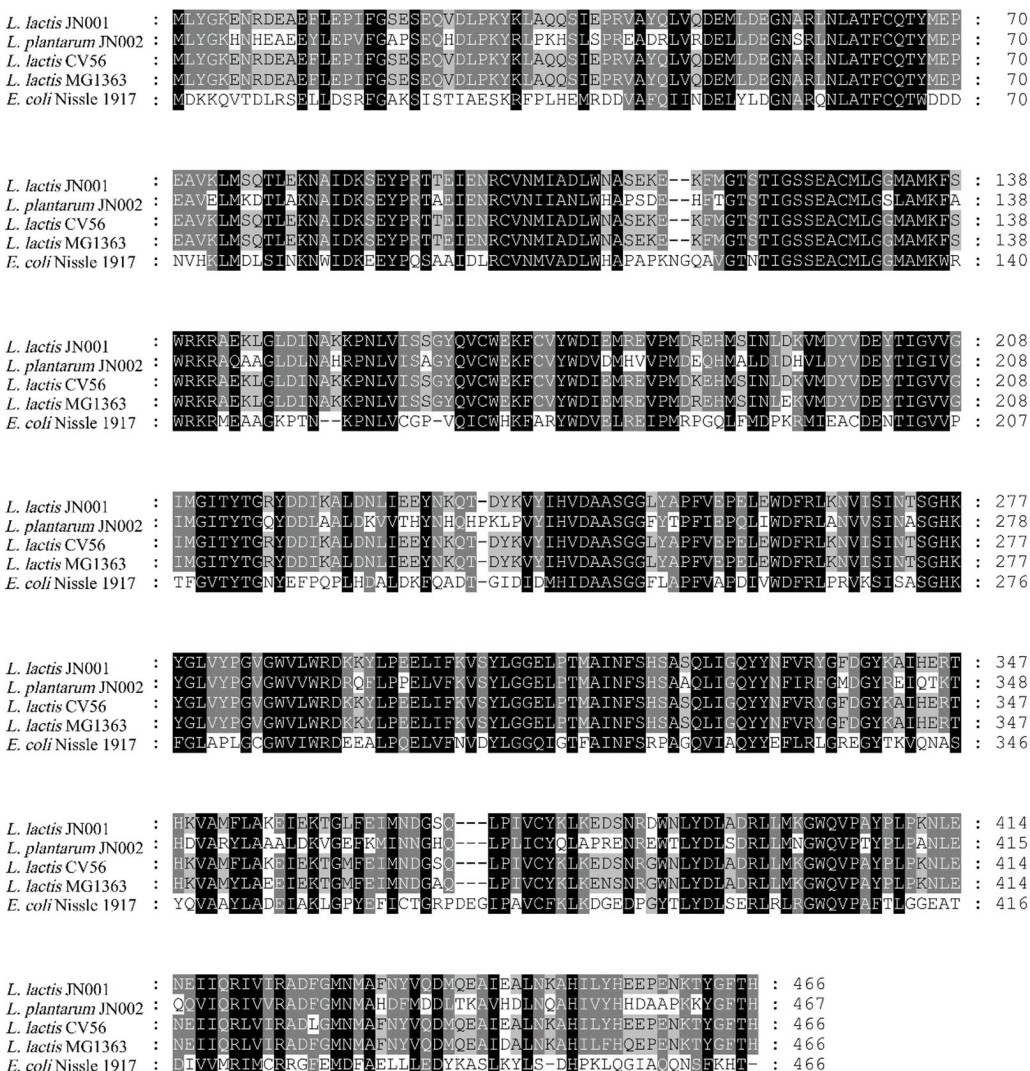

**Figure 1.** Alignment of the amino-acid sequences of selected GADs. Sequences were analyzed by MEGA6. The amino-acid sequences of GAD from *L. lactis* JN001 and *L. plantarum* JN001 were aligned with those of other GADs; *L. lactis* subsp. *lactis* CV56 (ADZ63898.1), *L. lactis* subsp. *lactis cremoris* MG1363 (CAL97772.1), and *E. coli* Nissle 1917 (UJL93810.1).

### 3.2. Production of GABA by Engineered C. tyrobutyricum

GABA production of recombinant strains was evaluated in 50 mL serum bottles (Figure 3). After 36 h of culture, a higher $OD_{600}$ was achieved by Ct-pMA02 and Ct-pMB01, and MSG was rapidly consumed by Ct-pMA02 in the first 24 h with a GABA production of 6.05 g/L. Moreover, a longer fermentation period was observed for Ct-pMA01 and Ct-pMB01 to reach the highest GABA titer, which was 5.76 g/L and 5.42 g/L, respectively. A slight decrease in GABA concentration was observed during the fermentation period of Ct-pMA02 after 48 h, which might have been due to GabT, gamma-aminobutyrate aminotransferase, which is responsible for GABA degradation, and the encoding gene was found in the *C. tyrobutyricum* genome. Similarly, degradation of GABA was also found in a previous report on *E. coli* [35]. On the contrary, no significant GABA degradation was observed for Ct-pMA01 or Ct-pMB01, which might have been due to the lower productivity of GABA compared to Ct-pMA02, indicating that GABA degradation might be induced by a high productivity of GABA. Overall, Ct-pMA02 exhibited the best GABA-synthesis ability.

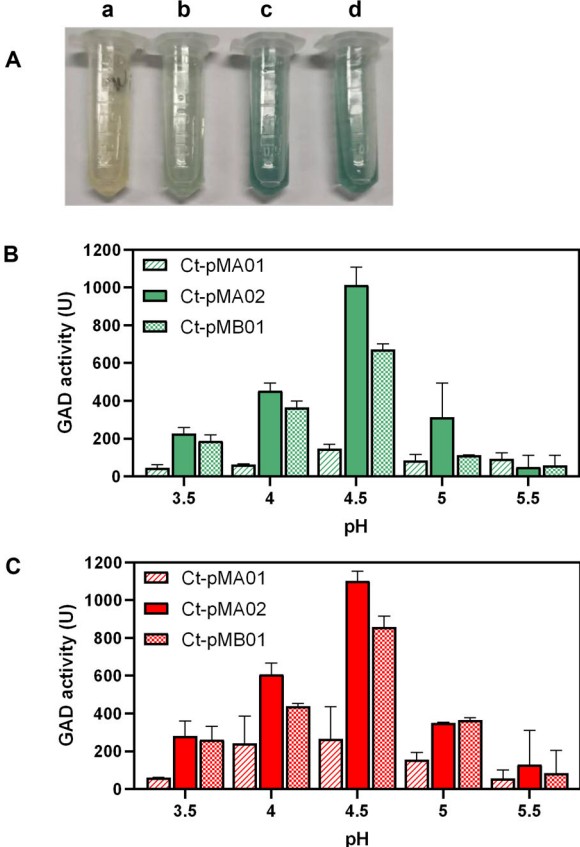

**Figure 2.** (**A**) The GAD activity of wild-type (**a**), Ct-pMA01 (**b**), Ct-pMA02 (**c**), and Ct-pMB01 (**d**) was determined by detecting the amount of GABA produced from MSG by the Berthelot method. (**B**) The GAD activity of cell extracts in the exponential phase. (**C**) The GAD activity of cell extracts in the stationary phase.

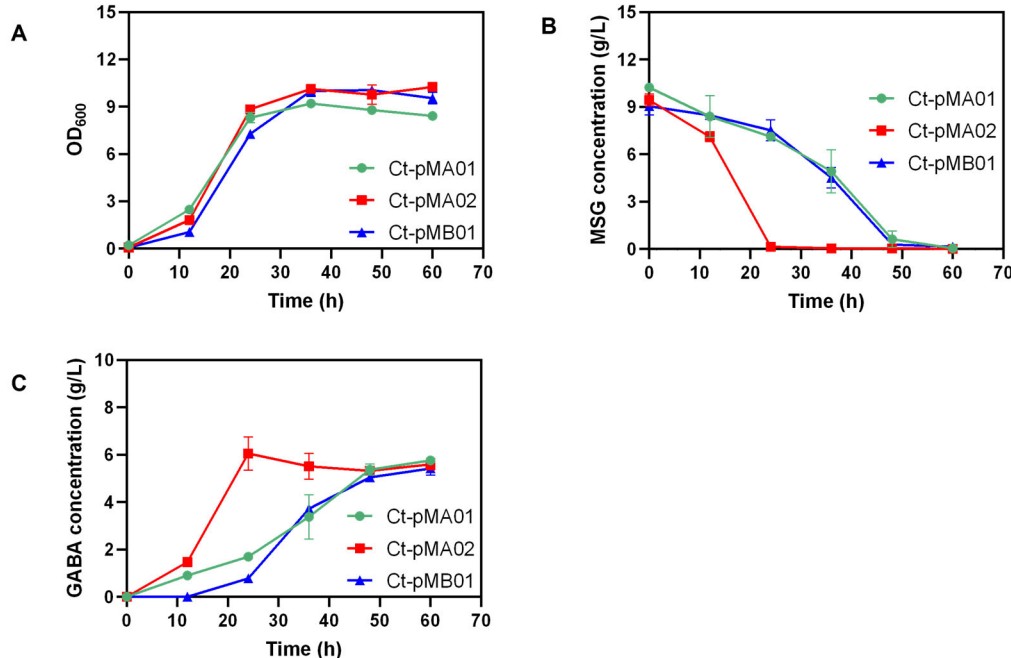

**Figure 3.** Batch fermentation in serum bottle by Ct-pMA01, Ct-pMA02, and Ct-pMB01. (**A**) $OD_{600}$; (**B**) MSG concentration; (**C**) GABA concentration. GABA synthesis of engineered *C. tyrobutyricum* strains was investigated in CGM medium containing 10 g/L MSG during 60 h of incubation at 37 °C.

### 3.3. Enhanced MSG Tolerance and GABA Production by Class I Heat-Shock Protein GroESL Overexpression

Quantitative real-time PCRs were carried out to test whether MSG stress could induce transcriptional changes of the Class I heat-shock genes (HSGs) in Ct-pMA02, and the results indicate that all six measured genes (*grpE*, *dnaK*, *dnaJ*, *groES*, *groEL*, and *htpG*) were highly overexpressed during the time course of all the MSG stress conditions tested (Figure 4A,B). The expression of *grpE*, *dnaK*, *dnaJ*, *groES*, *groEL*, and *htpG* increased by more than 1.5 times when treated with 10 g/L MSG and further increased with 40 g/L MSG, especially *grpE*, *groES*, *groEL*, and *htpG* (Figure 4A). With the prolonged treatment time, the expression of *grpE*, *dnaK*, *dnaJ*, *groES*, *groEL*, and *htpG* slowly decreased (Figure 4B). These results suggest that the expression of Class I heat-shock protein genes would be significantly up-regulated under stress from a high concentration of MSG, thus facilitating *C. tyrobutyricum* to respond to MSG stress. After adapting to MSG stress, the expression of these genes would gradually return to the normal level.

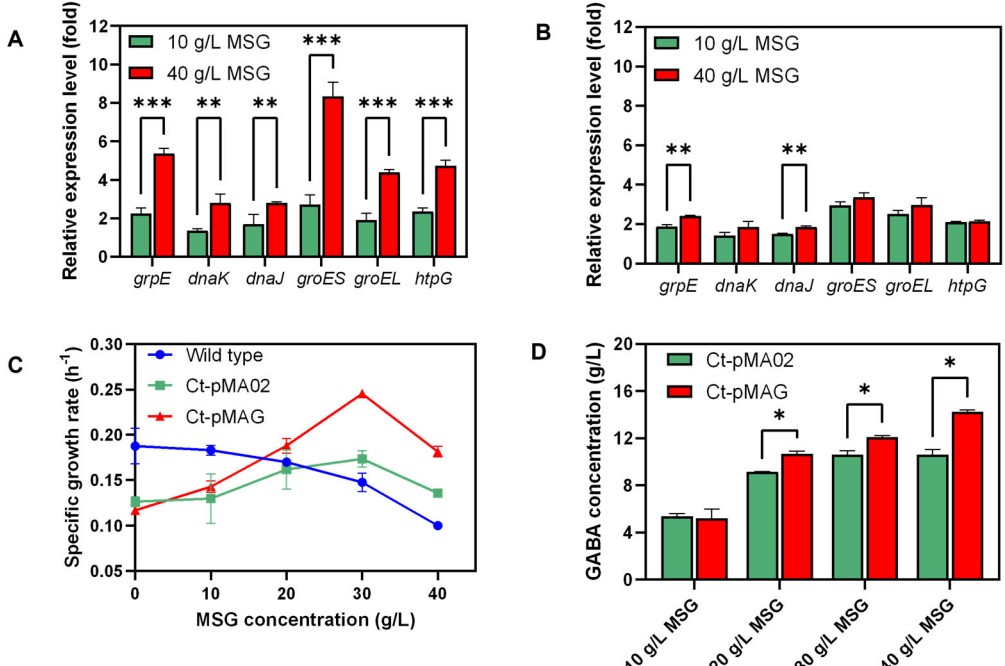

**Figure 4.** Relative expression level of *grpE*, *dnaK*, *dnaJ*, *groES*, *groEL*, and *htpG* under different concentration of MSG in Ct-pMA02. The error bars indicate the variations of three biological replicates. (**A**) MSG treatment for 15 min; (**B**) MSG treatment for 30 min. (**C**) The specific growth rate of wild-type, Ct-pMA02, and Ct-pMAG in CGM medium with different MSG concentrations (0–40 g/L) was compared. (**D**) Final GABA production of Ct-pMA02 and Ct-pMAG in CGM medium with different MSG concentrations (10–40 g/L) at 37 °C after 72 h. Statistically significant differences (*** $p < 0.001$, ** $p < 0.01$, * $p < 0.05$, Student's *t*-test) are indicated by asterisks.

Distinct increased expression levels for *groES* and *groEL* were observed under MSG stress, and the relative expression levels of *groES* and *groEL* were nearly 1.8 and 1.3 times higher, respectively, with the increased MSG concentration after treatment for 15 min (Figure 4A). Inspired by these facts, *groES* and *groEL* were overexpressed to test whether engineered strains could be more tolerant against MSG stress and improve GABA production. Based on the plasmid pMA02, *groES* and *groEL* were ligated to Pcat1 under *lcgadB* to generate plasmid pMAG, which was further transformed to *C. tyrobutyricum* to generate Ct-pMAG.

Specific growth rate was used to evaluate the inhibitory effect caused by MSG. Wild-type, Ct-pMA02, and Ct-pMAG strains were cultured with different concentrations of

MSG in 50 mL serum bottles to examine whether overexpression of HSGs could improve tolerance to MSG. The specific growth rate of the wild-type strain decreased gradually with increased MSG concentration; however, the specific growth rate of Ct-pMA02 and Ct-pMAG increased from 0 to 30 g/L and then dropped (Figure 4C). The specific growth rates of Ct-pMAG and Ct-pMA02 were highest when 30 g/L MSG were added. Ct-pMAG exhibited a generally higher specific growth rate under the MSG gradient, indicating that the tolerance to MSG stress was improved by HSG overexpression. The final GABA titer was measured as shown in Figure 4D. As for Ct-pMAG and Ct-pMA02, GABA production was similar under 10 g/L MSG; however, a higher titer was achieved by Ct-pMAG as the MSG concentration increased from 20 to 40 g/L. The highest GABA production (14.26 g/L) was observed under 40 g/L MSG using Ct-pMAG, which was 34.53% higher than that of Ct-pMA02. These results suggest that overexpression of *groESL* could improve GABA production, which is consistent with a previous report on *E. coli* [16].

### 3.4. Production of GABA in Fed-Batch Fermentation

As described above, strain Ct-pMAG exhibited the highest GABA production under the given condition. Thus, Ct-pMAG was selected for GABA synthesis in a 5 L bioreactor with pH-free and pH-control strategies. During the fermentation, *C. tyrobutyricum* produced butyric acid and acetic acid, leading to the acidic condition [36], which drove GABA synthesis. No additional alkalis needed to be added during the fermentation since GABA production consumed MSG and protons in the broth to maintain pH. When applying the pH-free strategy, Ct-pMAG grew rapidly within 12–36 h, and the pH of the fermentation broth dropped quickly to drive the synthesis of GABA. Consumption of protons resulted in an increase in pH. The MSG-consumption rate gradually decreased after fermentation for 60–84 h, resulting in a final GABA titer of 35.57 g/L (Figure 5A).

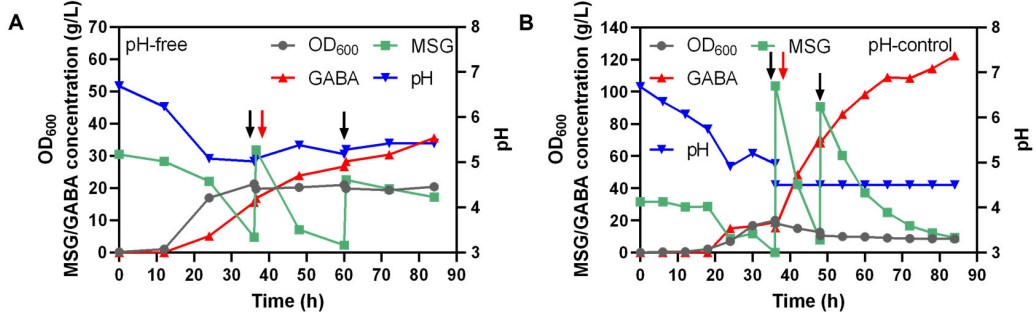

**Figure 5.** Fed-batch fermentation of Ct-pMAG in a 5 L bioreactor at 37 °C for 84 h. MSG and glucose were supplied during the period. (**A**) Fermentation using the pH-free strategy, with 2 times 50 mL MSG aliquot (600 g/L) added. (**B**) Fermentation using the pH-control strategy, with 2 times 200 mL MSG aliquot (700 g/L) added. The black arrow indicates the MSG-feeding moment and the red arrow indicates the 60 g/L glucose-feeding moment.

The pH-control strategy was applied to further improve the GABA production (Figure 5B). After the first 36 h, the fermentation process using the pH-control strategy was essentially the same as that under the pH-free strategy. Then, when the biomass peaked at 36 h, $H_2SO_4$ solution was supplied to maintain the pH at 4.5 to reach the highest GAD activity. A total of 200 mL MSG aliquot (700 g/L) was added to the bioreactor at 36 h and 48 h. After 48 h, the MSG-consumption rate and GABA synthesis steadily decreased, which might have been due to the substantial fall in biomass that was responsible for bio-transformation. At the end of the fermentation, a 122.34 g/L final titer of GABA was achieved by applying pH-control strategy, which was 3.43 times as much as that under the pH-free strategy (Figure 5A,B).

### 3.5. Improvement of GABA Biosynthesis by Two-Stage Strategy Using Glutamate Acid (L-Glu) as Substrate

As mentioned above, excessive soluble MSG inhibited cell growth. Although the detrimental effects of excessive MSG could have been decreased by using the fed-batch approach, continuous $H_2SO_4$ supplementation was required to achieve a high titer of GABA, which made the fermentation process more complicated. It was reported that GABA production could be further improved using L-Glu as the substrate instead of MSG [37,38]. L-Glu functioned equally as MSG during the glutamate decarboxylation. However, L-Glu was barely soluble in water (1.51 g/100 g water, 40 °C) and the pH value of the saturated water solution was about 3.2, which might have provided little inhibitory effect and ideal acidic conditions for the reaction. Thus, the supplement of L-Glu was equivalent to a substrate-sustained release process, as the solution of L-Glu was accompanied with substrate consumption by glutamate decarboxylation dynamically.

Compared to Ct-pMA02, Ct-pMAG also showed improved GABA production during the bioconversion of L-Glu (Figure S3). To further enhance GABA production in the 5 L bioreactor, a two-stage strategy was applied, including a growth stage (Figure 6A–C) and a bioconversion stage (Figure 6D–F). Totals of 60, 90, and 120 g/L glucose were supplied to examine how the carbon source would influence the cell growth and GABA synthesis. Additionally, $NH_3 \cdot H_2O$ was supplied at the growth stage to maintain the pH at 6 for cell growth. The biomass of Ct-pMAG with 60 g/L glucose increased quickly after the first 16 h and peaked at 24 h, reaching an $OD_{600}$ of 26.4. When 90 g/L and 120 g/L glucose were added initially, the strain grew quickly after 16 h of culture and obtained a higher $OD_{600}$ than that with 60 g/L glucose at the end of the growth stage (Figure 6A). Cell-bound activity was also examined during the growth stage and the results indicate that the maximal cell-bound activity in the three groups was comparable (Figure 6B); however, the addition of 90 g/L glucose resulted in the highest total cell-bound activity (Figure 6C). At the beginning of the bioconversion stage, 588.52 g/L L-Glu was supplied in the bioreactor. Under the condition of 60 g/L glucose, 277.96 g/L GABA were produced with a productivity of 16.35 g/L/h after bioconversion for 17 h. The GABA production and productivity were improved under the condition of 90 g/L and 120 g/L glucose, which were 400.32 g/L and 36.39 g/L/h, and 400.55 g/L and 36.41 g/L/h, respectively (Table 1).

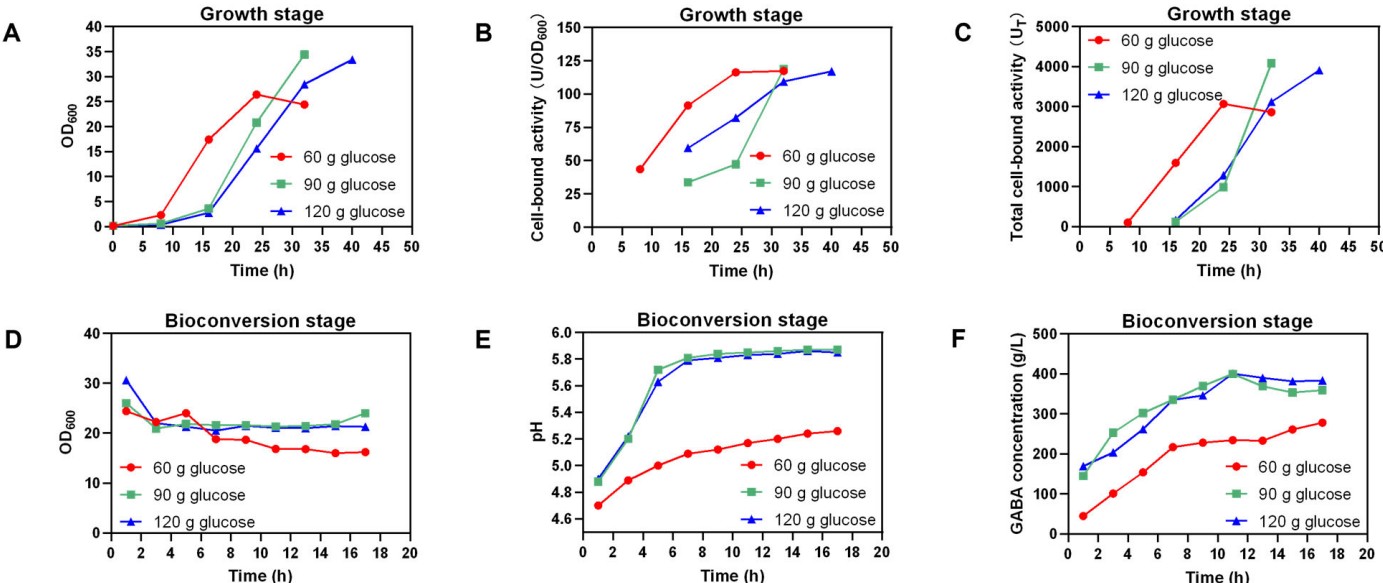

**Figure 6.** Profiles of two-stage bioconversion of L-Glu into GABA by Ct-pMAG. $OD_{600}$, cell-bound activity, and total cell-bound activity in the growth stage are shown in (**A–C**), respectively. $OD_{600}$, pH, and GABA concentration in the bioconversion stage are shown in (**D–F**), respectively. Total cell-bound activity ($U_T$) = $OD_{600} \times$ cell-bound activity.

**Table 1.** Comparison of GABA production from L-Glu by Ct-pMAG under the two-stage strategy.

| Sugar Concentration (g/L) | Initial OD$_{600}$ of Bioconversion Stage | Titer (g/L) | Productivity (g/L/h) | Molar Yield (%) |
|---|---|---|---|---|
| 60 | 24.40 | 277.96 | 5.67/16.35 * | 67.40 |
| 90 | 34.40 | 400.32 | 9.31/36.39 * | 97.07 |
| 120 | 33.4 | 400.55 | 7.85/36.41 * | 97.13 |

* Indicates that the productivity was calculated only considering bioconversion time, and the culture time of the bacteria was not included.

## 4. Discussion

As a major inhibitory neurotransmitter in the mammalian central nervous system, GABA has been employed in various fields. A cost-effective method of GABA biosynthesis was required to meet the growing needs in the market. *C. tyrobutyricum* ATCC 25755 is a non-pathogenic strain and could attain a large amount of biomass under anaerobic conditions, which is necessary in the industry application [26,29,36]. Compared to employing pure GAD enzymes, using cells as a biocatalyst to generate GABA is a promising method because of its efficiency and stability. Several studies have been conducted on metabolic engineering of GABA-producing bacteria, focusing on improving GABA-production titer and productivity, which are summarized in Table 2.

**Table 2.** Summary of strategies for enhanced GABA production using wild-type and metabolically engineered strains.

| Microorganism | Engineering/Process Strategy | Substrate | Operation Mode and Strategy | Titer (g/L) | Productivity (g/L/h) | References |
|---|---|---|---|---|---|---|
| *E. coli* BW25113 | ↓*lacI*, ↓*gabT*, ↓*sucA*, ↓*aceA* (IPTG based dynamic control) | Glucose | Batch, aerobic | 4.8 | 0.15 | [9] |
| *E. coli* BW25113 | +*Lactococcus lactis* GAD mutation, removing C-plug of GadC, ↑*groESL*, Δ*gabT* | L-Glu | Batch, bioconversion, aerobic | 308.26 | 44.04 * | [16] |
| *Lactobacillus brevis* NCL912 | Wild type | L-Glu | Batch, aerobic | 205 | 4.8 | [38] |
| *Lb. brevis* CGMCC1306 | ΔF$_0$F$_1$-ATPase | MSG | Batch, pH-control, aerobic | 43.65 | 0.9 | [39] |
| *L. brevis* ATCC367 | Δ*glnR* | MSG | Fed-batch, pH-control, aerobic | 284.7 | 3.95 | [12] |
| *Bifidobacterium. adolescentis* JCM 1275 | ↑*gadBC* | MSG | Fed-batch, pH-control, anaerobic | 43.99 | 0.73 | [19] |
| *C. glutamicum* | +*L. plantarum* GAD, Δ*dapA*, Δ*argB*, Δ*proA* | Glucose | Fed-batch, pH-control, aerobic | 70.6 | 1 | [8] |
| *C. tyrobutyricum* ATCC25755 | +*L. lactis* GAD, ↑*groESL* | MSG | Fed-batch, pH-free, anaerobic | 35.57 | 0.42 | This work |
| *C. tyrobutyricum* ATCC25755 | +*L. lactis* GAD, ↑*groESL* | MSG | Fed-batch, pH-control, anaerobic | 122.34 | 1.46 | This work |
| *C. tyrobutyricum* ATCC25755 | +*L. lactis* GAD, ↑*groESL* | L-Glu | Fed-batch, two stage, anerobic | 400.32 | 9.31/36.39 * | This work |

* Indicates that the productivity was calculated only considering bioconversion time, and the culture time of the bacteria was not included. ↓ and ↑ indicated down regulate and up regulate, respectively. + indicated heterologous expression.

GAD played a key role in the GABA synthesis; however, *C. tyrobutyricum* ATCC 25755 could not synthesize GABA due to a lack of native GAD. GAD from *E. coli* (EcgadB) is one of the most widely studied enzymes, and its crystal structure has been reported [9,40]. LcgadB was reported to have a low *Km* value that possesses a potentially high affinity with the substrate and high activity [18,41,42]. Recently, many studies on GAD from the application of *L. plantarum* (LpgadB) have been carried out due to its relative wide pH and activity [11,14]. Thus, three GADs were introduced in *C. tyrobutyricum*, and the GAD activity of LcgadB was the highest, followed by LpgadB. GAD expression was analyzed by SDS-PAGE, and the results show that there were deeper bands for Ct-pMA02 (LcgadB) and Ct-pMB01 (LpgadB) near the 55 kDa marker, as predicted (Figure S2). However, the expected band for Ct-pMA01 (EcgadB) was unclear, indicating a low expression level of EcgadB.

During GABA fermentation, bacteria constantly encounter stress caused by a high concentration of MSG as a substrate, which has a side effect on bacterial growth and GABA production [13]. It was reported that heat-shock proteins are multifunctional in response to extracellular stress. In *Clostridium botulinum*, heat-shock proteins were reported to play a significant role against pH and NaCl [43]. Butyric-acid tolerance and production were improved by overexpressing *groES* and *groEL* in *C. tyrobutyricum* [29]. The increased relative-expression level of *groES* and *groEL* indicate that the bacterial cells were sensitive in response to MSG stress (Figure 4A). Thus, *groES* and *groEL* were overexpressed to investigate the potential for enhanced GABA production. Compared to wild-type, the specific growth rate of Ct-pMA02 and Ct-pMAG increased over the range of 10–30 g/L and then steadily decreased. This might have been due to GABA synthesis consuming protons, creating a more alkaline environment suitable for bacterial growth [12]; however, an excessive amount of MSG (>30 g/L) inhibited the bacterial growth. Improved GABA production was observed when 20–40 g/L MSG was added using Ct-pMAG (overexpressed *groES* and *groEL*) (Figure 4D), indicating that GroESL might help strains against low pH and MSG stress to grow better under given conditions (Figure 4C). Although no significant difference was observed in cytoplasmic GAD activity between Ct-pMAG and Ct-pMA02 (Figure S4A), a higher cell-bound activity was achieved by Ct-pMAG (Figure S4B), which is consistent with previous work on *Lactobacillus brevis* [33].

When using the pH-free strategy in fed-batch fermentation, an excessive amount of MSG remained, which was possibly due to the low GAD activity caused by the increased pH of the fermentation broth (Figure 5A). Moreover, the final titer of GABA (34.87 g/L) did not increase with more MSG supplied (Figure S5A). Previous reports revealed that relative acidic conditions were essential for the reaction of glutamate decarboxylase [8,14]. The pH-control strategy is common in microbial GABA synthesis [2]. By applying the pH-control strategy, the pH was adjusted to 4.5 to attain the highest GAD activity to generate GABA synthesis, and increase in the GABA titer (62.59 g/L) was observed under equivalent MSG supplementation (Figure S5B). With more MSG supplied, a higher titer (122.34 g/L) and productivity (1.46 g/L/h) were achieved (Figure 5B). Many studies have revealed that L-Glu as a substrate could enhance GABA synthesis compared with MSG as a substrate [37,38]. In this study, L-Glu was also investigated as a substrate. However, during fermentation with 10 g/L L-Glu added, wild-type and engineered strains Ct-pMA01, Ct-pMA02, Ct-pMAB01, and Ct-pMAG all failed to grow (data not shown). The pH was considered to be the major obstacle since the initial pH was nearly 3.2, which was not suitable for the growth of *C. tyrobutyricum*. It was reported that *L. brevis* NCL912 could grow well in an extremely-low-pH environment and synthesize GABA, and the initial pH of the fermentation medium with L-Glu added was approximate 3.3 [38]. L-Glu inhibited the growth of *C. tyrobutyricum* when it was added to the medium at the beginning of the fermentation, but it could also be added once there are enough cells (as biocatalyst) to catalyze the synthesis of GABA. Using the two-stage strategy, firstly, the fermentation procedure began with the accumulation of biomass. At the bioconversion stage, a higher

GABA titer (400.32 g/L) and productivity (36.39 g/L/h) were achieved under the condition of 90 g/L glucose with a competitive molar yield (97.07%) (Figure 6F).

**5. Conclusions**

In conclusion, this study suggested that *C. tyrobutyricum* could synthesize GABA by introducing GAD gene. By co-expressing Class I heat-shock genes *groES* and *groEL*, strain tolerance against MSG was improved and a higher GABA titer was achieved. In fed-batch fermentation using MSG, GABA production reached 35.57 g/L and 122.34 g/L using the pH-free strategy and pH-control strategy, respectively. To further improve GABA production, a two-stage strategy was carried out, achieving a final GABA titer of 400.32 g/L with a productivity of 36.39 g/L/h from L-Glu. Thus, our study exhibits a promising method and platform for high-level GABA production using *C. tyrobutyricum*.

**Supplementary Materials:** The following supporting information can be downloaded at: https://www.mdpi.com/article/10.3390/fermentation9050445/s1, Table S1. Primers used in this study; Table S2. Strains and plasmids used in this study [31,44]; Figure S1. Electrophoresis results of colony PCR for recombinant strains; Figure S2. SDS-PAGE of cell extracts prepared by wild-type and engineered *C. tyrobutyricum* strains; Figure S3. Profiles of bioconversion of L-Glu using Ct-pMA02 and Ct-pMAG; Figure S4. (A) Comparison of GAD activity between Ct-pMA02 and Ct-pMAG; Figure S5. Fed-batch fermentation of Ct-pMAG in a 5 L bioreactor at 37 °C for 84 h.

**Author Contributions:** Z.L.: Conceptualization, methodology, investigation, formal analysis, data curation, original draft preparation; X.G.: methodology, investigation, writing—reviewing and editing; K.D.: investigation, writing—review and editing; J.F.: investigation, writing—review and editing; T.Z.: investigation, writing—review and editing; H.F.: conceptualization, methodology, investigation, resources, writing—review and editing; J.W.: writing—review and editing, supervision, project administration, funding acquisition. All authors have read and agreed to the published version of the manuscript.

**Funding:** This work was supported by the National Natural Science Foundation of China (22178133, and 21808069).

**Institutional Review Board Statement:** Not applicable.

**Informed Consent Statement:** Not applicable.

**Data Availability Statement:** All data generated or analyzed during this study are included in this published article and its Supplementary Materials.

**Conflicts of Interest:** The authors declare no conflict of interest.

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
