# Peer review of "Biosynthesis of Gamma-Aminobutyric Acid by Engineered Clostridium tyrobutyricum Co-Overexpressing Glutamate Decarboxylase and Class I Heat Shock Protein"

_fermentation, doi:10.3390/fermentation9050445_

Round 1

Reviewer 1 Report

Please see the review in the attached file.

Reviewer 2 Report

This study proposed a promising method for high-level production of GABA using C. tyrobutyricum by introducing the GAD gene. The tolerance of the strain against MSG was improved, and higher GABA titers were achieved by co-expressing Class I heat shock genes groES and groEL. A two-stage strategy was then employed, resulting in a final titer of 400.32 g/L of GABA with L-Glu as a substrate. These works have provided very helpful assistance for the industrial production of GABA.

There are some minor mistakes that should be improved.

  1. P3 line 23: Please provide a brief summary of the experimental procedures.
  2. P4 line 178: What is "acid water body"?
  3. P6 line 226: "E. coli" should be in italics.
  4. Section 3.4: There may be comparability issues in the results conducted with different amounts of MSG added in pH-free and pH-control experiments. To ensure accurate comparison, it is recommended to conduct experiments under the same conditions.
